# Behavioural Aversion and Cortisol Level Assessment When Adult Zebrafish Are Exposed to Different Anaesthetics

**DOI:** 10.3390/biology11101433

**Published:** 2022-09-30

**Authors:** Jorge M. Ferreira, Sara Jorge, Luís Félix, Gabriela M. Morello, I. Anna S. Olsson, Ana M. Valentim

**Affiliations:** 1i3S—Instituto de Investigação e Inovação em Saúde, Universidade do Porto, 4200-135 Porto, Portugal; 2IBMC—Instituto de Biologia Molecular e Celular, Universidade do Porto, 4200-135 Porto, Portugal; 3ICBAS-UP—Instituto de Ciências Biomédicas Abel Salazar, Universidade do Porto, 4050-313 Porto, Portugal; 4FCUP—Faculdade de Ciências da Universidade do Porto, Universidade do Porto, 4169-007 Porto, Portugal; 5CIIMAR—Centro Interdisciplinar de Investigação Marinha e Ambiental, 4450-001 Matosinhos, Portugal; 6CITAB—Centro de Investigação e de Tecnologias Agroambientais e Biológicas, Universidade de Trás-os-Montes e Alto Douro, 5001-801 Vila Real, Portugal; 7CHH—Common Home of Humanity, 4430-211 Vila Nova de Gaia, Portugal

**Keywords:** anaesthesia, refinement, aversion, cortisol, zebrafish

## Abstract

**Simple Summary:**

The increased popularity of zebrafish as a research model calls for appropriate refinement of procedures, such as finding the best anaesthetic protocol to use in adult zebrafish in a laboratory setting. The scarce literature available regarding aversion described the most used anaesthetic (MS222) as aversive to adult zebrafish; therefore, the study of alternatives is urgent. Thus, we studied the impact of three anaesthetic alternatives (a combination of propofol with lidocaine, clove oil, and etomidate), and the standard anaesthetic (MS222) on zebrafish aversion and on cortisol levels. Contrary to what was expected, MS222 did not result in clear signs of aversion, while only etomidate generated a similar profile to the aversive substance (hydrochloric acid). Our results suggested that all the anaesthetic protocols except for etomidate were valid candidates for use in a laboratory setting, although none were innocuous.

**Abstract:**

The use of zebrafish (*Danio rerio*) as an animal model is growing and occurs in a wide range of scientific areas. Therefore, researchers need better and more appropriate anaesthetics for stressful and/or painful procedures to prevent unpleasant experiences. Thus, we aimed to study if adult zebrafish displayed aversion-associated behaviours (conditioned place aversion) and alterations in cortisol levels when exposed to equipotent concentrations of MS222, propofol/lidocaine, clove oil, or etomidate. Adult AB zebrafish (mixed-sex, N = 177) were randomly assigned to MS222 (150 mg/L), Propofol/Lidocaine (5 mg/L propofol + 150 mg/L lidocaine), Clove Oil (45 mg/L), or Etomidate (2 mg/L) groups. The conditioned place aversion test was used to assess behavioural aversion. Only etomidate resulted in a similar aversion to the positive control group (HCl; pH = 3). Cortisol levels were measured 5 and 15 min after loss of equilibrium. Etomidate induced low levels of cortisol by impairing its synthesis, whereas all the other groups had similar cortisol levels. Based on our data, etomidate was ruled out as an alternative to MS222, as it showed an aversive profile. The remaining protocols were not innocuous, displaying a weak aversive profile when compared to the positive control. In conclusion, a combination of propofol with lidocaine, clove oil, and MS222 were valid candidates for use as anaesthetic protocols.

## 1. Introduction

According to the latest European Commission report on the statistics on the use of animals for scientific purposes in the EU [1], fish are the second most used animal group in research (13%); with zebrafish representing 40% of these. The refining of procedures is essential for responsible high-quality research. As several procedures involve stress and/or pain, the use of anaesthesia is often required. Most studies only evaluate the efficacy of anaesthetic protocols in terms of their capacity to induce anaesthesia that is adequate for the procedure, providing immobility and/or analgesia, and preventing physical harm and distress. However, the potential aversive effect of anaesthesia is poorly studied, raising welfare concerns. In fact, only two studies [2,3] were performed to study this issue in zebrafish, and more data are needed to evaluate which protocol fully respects zebrafish welfare.

MS222 or tricaine has been shown to be aversive to adult zebrafish, and this aversion was independent of its acidic profile, as animals showed signs of aversion or distress regardless of the anaesthetic solution being buffered or not [2,3,4,5]. Nevertheless, MS222 is the standard and most used anaesthetic in zebrafish research [6], probably due to its traditional use in larger fish species, as it is the only anaesthetic approved by the Food and Drug Administration agency as safe to use in the fish industry. Considering the risk of causing aversion, the development of alternative anaesthetic protocols to provide better animal welfare and produce more reliable scientific data is needed. Previous studies suggested the combination of propofol with lidocaine, clove oil, and etomidate as being promising alternatives [2,3,7].

Propofol, a short-acting sedative-hypnotic, consistently induces analgesia in zebrafish but only at high concentrations [8], according to recent studies [9], which may raise the danger of overdosing. Other studies have shown that immersing zebrafish in a water bath containing a high concentration of the local anaesthetic lidocaine had analgesic and/or anaesthetic effects [10,11], but this also increased zebrafish mortality [10]. However, when combining propofol with lidocaine, the dose of each individual anaesthetic can be decreased [8,12,13], inducing a balanced and safer anaesthesia. Another anaesthetic used is clove oil, which is highly lipophilic and rapidly absorbed through the skin and gills, entering the bloodstream, crossing the blood–brain barrier, and inducing loss of consciousness. Clove oil has been demonstrated to minimise cortisol response compared with MS222 [14,15]. This agent is widely available, has a low cost, and has a wide range of efficacy (i.e., from 2 to 5 μg/mL for sedation to 60 to 100 μg/mL for immersion anaesthesia) [16], but it needs to be solubilised in an organic solvent such as ethanol prior to anaesthesia, which increases the workload while introducing one more chemical compound [17,18]. Another potential downside is that clove oil contains low levels of methyleugenol, which is considered carcinogenic [19]. Contrarily, etomidate is a safe nonbarbiturate hypnotic agent that has been described to block cortisol [20,21], the same effect its derivative metomidate has in fish [14,22]. Etomidate does not provide analgesia, but it is a relatively fast-acting anaesthetic agent [12].

In order to improve anaesthesia for adult zebrafish, the study’s primary goal was to identify the least aversive anaesthetic protocol when testing the MS222, propofol/lidocaine, clove oil, and etomidate anaesthetics. The conditioned place aversion test (CPA) and cortisol level quantification were used to assess zebrafish behavioural aversion and physiological stress.

## 2. Materials and Methods

### 2.1. Ethics Statement

All procedures were carried out under personal and project licenses approved by the National Competent Authority for animal research (Direção-Geral de Alimentação e Veterinária, Lisbon, Portugal) (approval number: 014703) and by the Animal Welfare and Ethics Review Body of the Institute for Research and Innovation in Health (i3S). The European Directive 2010/63/EU on the protection of animals used for scientific reasons, as well as its transposition into Portuguese law (“Decreto Lei” 113/2013), were followed in all experimental procedures.

### 2.2. Animals and Housing

Sample-size calculation was performed in G*Power 3.1 (University of Düsseldorf, Düsseldorf, Germany) while assuming a type II error probability of α = 0.05, a power of 0.85, and an effect size of 0.988 based on previous data [7]. Adult (7–16 months old) mixed-sex AB zebrafish (N = 177; 81 for CPA + 96 for cortisol) bred in the Animal Facility of i3S were used. They were kept in groups of eight fish in 3.5 L tanks on a 14 h:10 h light:dark cycle in a recirculating water system connected to a central unit of water purification under controlled conditions (27.0 ± 0.5 °C, pH of 7.0 ± 0.5, and conductivity of 700–715 S). Fish were fed thrice daily with a commercial diet (Zebrafeed 400–600 µm, Sparos, Olho, Portugal). The same water and light:dark cycle conditions were maintained in the conditioned place aversion apparatus where the animals were housed during the experiment.

### 2.3. Anaesthetic Solutions Tested

For this study, we used 69 mixed-sex zebrafish (57% male to 43% female) randomly divided into six groups: HCl (animals subjected to water with a pH of 3 induced by hydrochloric acid; n = 11), MS222 (Sigma-Aldrich, Burlington, MA, USA) (n = 12), Propofol/Lidocaine (P/L) (propofol was Lipuro 2%, B. Braun Melsungen AG, Germany, and 2%, Braun, Queluz de Baixo, Barcarena, Portugal) (n = 12), Clove Oil (Merck, Kenilworth, NJ, USA) (n = 12), Etomidate (Lipuro 2 mg/mL, B. Braun, Queluz de Baixo, Barcarena, Portugal) (n = 10), and Vehicle (Intralipid, Sigma-Aldrich, Burlington, MA, USA) (n = 12); the concentrations of the compounds used in these groups are displayed in Table 1. The pH of all solutions was around 7 (6.97 ± 0.11) (Table 1). The animals from the anaesthetic groups were exposed to equipotent concentrations to induce loss of equilibrium that were established prior to the study [7]. All anaesthetic solutions were freshly prepared except the MS222 and clove oil. The MS222 test solution was prepared from a buffered (pH = 7) stock solution (10 g/L) and the clove oil test solution from a stock solution of 10% clove oil and 90% ethanol.

Hydrochloric acid at a pH of 3 is aversive to zebrafish [3], thus a HCl group was used as a positive control to validate the behavioural methodology used to study aversion. As propofol and etomidate have a white colour, the fish could react to visual cues and not to the actual exposure to the anaesthetics. To rule out this influence, a group exposed only to the common vehicle of these compounds (Intralipid) was added as a positive control for the white colour—Vehicle group. For the P/L and the Etomidate groups, during the Post-exposure phase, the Intralipid vehicle was added to the water at the same concentration as was present in the anaesthetics, thus mimicking their colours. The Vehicle group had the same concentration of Intralipid as the propofol in the P/L group. The other testing solutions were transparent (Appendix A).

### 2.4. Conditioned Place Avoidance Test

To test behavioural aversion, we used a methodology adapted from Wong et al. [2] and Ferreira [7]. This experiment was carried out in an apparatus that had two opaque black aquariums connected by a tube so that the fish could travel between them; each tank had a small transparent window (Figure 1).

Each aquarium had a lid equipped with an LED light source, but only one of the LED lights was lit at a given time. Vertical plastic plants (from a pet store) were added as an environmental enrichment to minimise the possible negative impact of isolation [23]. Plants were always placed on the opposite wall of the tube on both sides of the apparatus. Aeration was additionally provided and was positioned near the plant. The conditioned place avoidance paradigm involved associating an aversive experience with a setting that was previously viewed as neutral or favourable, leading to the avoidance of the neutral or preferred location [22]. The degree of avoidance was proportionated to the time spent on each side of the apparatus and the latency to enter an environment that was previously perceived as neutral or positive [3] (Figure 2). The animal remained inside this apparatus for the entire conditioned place avoidance test, which had a maximum duration of 13 days.

#### 2.4.1. Habituation

This first phase, which lasted 48 h, involved placing each animal separately in the apparatus for acclimatisation. Animals were fed five times a day on the side where they were found during this phase. The animals were only routinely observed for health at this point without any testing being performed.

#### 2.4.2. Training Phase

Animals were moved to the lighted side of the apparatus during this phase before being fed by being gently pushed with a net in the direction of the tube; animals that were already on that side were also subjected to contact with the net without being exposed to the air. The placement of a transparent partition ensured that the animals remained on the lit side and were unable to cross over to the darkened side. Then the sides’ lighting was switched, food was added to the newly lit side, the partition was lifted, and the animals were free to move around and eat. The animals’ innate preferences for this environment were reinforced by feeding them on the lit side. Additionally, the animals were trained to pass through the tube after the removal of the divider and the light switch, which served as cues to indicate the presence of food on the other side. Five trials per day were used to measure the latency to enter the lit side during this phase following the light switch and partition removal. This phase lasted for up to 10 days or until the subjects met the criteria for being fully trained, which was when the subjects entered the lit side at least three times in a period of one minute or less and never took more than two minutes in the remaining trials on any given day for three consecutive days. The five latencies to enter the lighted side in the last day of training were averaged and designated as the training baseline, which was later compared with the latency of the testing phase.

#### 2.4.3. Testing Phase

Pre-exposure: a total of 24 h after being fully trained, animals were video-recorded for 15 min without any interference and the time spent in each side was noted.

Exposure: immediately after pre-exposure, the transparent barrier was placed when the fish was on the lighted side while the test solution and the food were introduced on the opposite side. At this point, the light conditions were switched between sides and the barrier was lifted. Fish were expected to quickly enter the lighted side, similarly to the training baseline. This trial ended when an animal spent 3 consecutive minutes on the side with the anaesthetic/test solution or when 15 min elapsed irrespective of the animal’s location. Three minutes was considered a safe amount of time to ensure that the animal was unconscious [13] and there was no risk of overdose; this was established because monitoring the fish was difficult due to the apparatus being opaque and having only a small window. After this trial, fish recovered in a tank with clean water from the system for at least 30 min; after this period, all animals exhibited normal swimming and behaviour. Animals that did not enter into contact with the test solution were not further evaluated. In this phase, only latency to enter the lighted side was recorded.

Post-exposure: following recovery, the animal was introduced again into the darkened side of the experimental apparatus with the barrier placed on this side for five min to prevent the fish from passing to the lighted side. Following the 5 min introduction period, the barrier was lifted and the animal was allowed to explore the entire apparatus for 15 min to verify if there was any evidence that the fish was conditioned to avoid the side on which it was previously exposed to the test solution. In this trial, no anaesthetic or HCl solution was added to the water. However, the P/L and etomidate groups were also exposed to the vehicle in this phase so that the visual cue (white colour) was present without the active compound of the anaesthetics (Appendix A). To evaluate the aversion caused by each anaesthetic, data on the length of time spent in each side, the number of visits, and the latency to enter the lighted side were recorded. The trial was over after 15 min regardless of whether the fish entered the lit side. In this case, a value of 900 s (maximum latency) was attributed to these animals regarding the latency to enter in the lighted side. All videos were recorded with a camera (Legria HF R606, Canon, Ōta, Japan) facing the tunnel and were later analysed with the use of Ethowatcher (Universidade Federal de Santa Catarina, Santa Catarina, Brazil) [23], a behavioural-event-logging software.

To ensure that there was no significant diffusion of the anaesthetic to the opposite side of the apparatus, a trial using food colouring and without any fish was conducted prior to the CPA. The minimal diffusion of this substance to the opposite side after 30 min (more than the trial’s duration) suggested that little to no testing solution—if any—entered the apparatus’s darkened side during the testing phase.

### 2.5. Cortisol Determination

Trunk cortisol levels have been assessed previously as an indication of stress response in smaller adult fishes such as zebrafish [24,25]. Another batch of zebrafish were euthanised by rapid cooling followed by decapitation 5 or 15 min after loss of equilibrium in response to the anaesthesia exposure (n = 12). If they were females, the eggs were gently removed by pressing their abdomen [26] and then each trunk was weighed and placed in a 15 mL centrifuge tube at −20 °C containing 5 mL of ice-cold phosphate-buffered saline (PBS, pH 7.4). Trunk samples were processed for cortisol measurement as previously described [27]. Samples were rapidly thawed on ice and homogenised twice in 500 μL of PBS using the automatic lyser TissueLyser II (Qiagen, Hilden, Germany) at 30 Hz for 90 s. Cortisol was extracted into 500 μL diethyl ether and vortexed for 30 s. Samples were further centrifuged at 5000 rpm (Heraeus Biofuge Pico, Hanau, Germany) for 10 min and frozen at −20 °C to collect the organic layer. The extraction was repeated twice and the final tubes containing the diethyl ether extract were dried at 45 °C under reduced pressure in a Labconco centrifugal evaporator (model: Centrivap 78120-00 D). The dried extracts were portioned with 500 μL of PBS and the same amount of hexane to prevent interference from the precipitated lipids. Samples were then vortexed and centrifuged as before. The aqueous phase was stored at −20 °C until analysis. For the analysis of cortisol levels, 50 μL of the reconstituted samples was used and assayed at 405 nm with a correction at 490 nm on a microplate spectrophotometer (PowerWave XS2, Bio-Tek Instruments, Winooski, VT, USA) by following the ELISA kit instructions (Salimetrics assay #1-3002; Salimetrics, State College, PA, USA). Values are presented in ng per gram of the animal body weight.

### 2.6. Data Analysis

Response variables (latency to enter the lighted side, time spent on the lighted side, cortisol levels, number of entries on the lighted side, and probability of entering the lighted side at least once) were tested as functions of the studied fixed and random effects, as detailed below. Different modelling approaches were used for each responsible variable to ensure a normal distribution of residuals and homoscedasticity. For all fitted models, least-square means were compared while considering a 95% confidence interval with Tukey–Kramer corrections for multiple pairwise comparisons. All the analyses were performed using SAS University Edition^®^ (SAS Institute Inc., Cary, NC, USA).

For analyzing latency to enter the lighted side, the GLIMMIX procedure was used with a log-normal response distribution for testing latency to reach the lighted side (response variable) as a function of the fixed-effect treatment group (six-level categorical variable with levels of HCL, Etomidate, Clove Oil, P/L, Vehicle, and MS222), fixed effect of phase (three-level categorical variable with levels of the training baseline, exposure, and post-exposure), and the interaction between these effects while considering fish as random effect.

For the analysis of time spent in the lighted area, an index was calculated by dividing the total time spent by the fish on the lighted side by the summation of the time spent on both sides; we excluded the time spent by the fish in the communicating tube because there was substantial variation among individual fishes in the total time spent in the tube. In this way, the calculated index was an indication of the proportion of time spent in the lighted area relative to the total time that the fish was in one of the areas. To achieve a normal distribution of model residuals and homoscedasticity, a square-root data transformation was performed on the calculated lighted-side index. The MIXED procedure was used for testing the square-root-transformed lighted-side index (response variable) as a function of the fixed-effect treatment group (six-level categorical variable with levels of HCL, Etomidate, Clove Oil, P/L, Vehicle, and MS222), fixed effect of phase (two-level categorical variable with levels of pre-exposure and post-exposure), and the interaction between these effects while considering fish as a random effect.

Two different approaches were used to evaluate fish entrance into the lighted side. First, the probability of fish entering the lighted side at least once was modelled via logistic regression by using the LOGISTIC procedure while considering the fixed-effect treatment group (six-level categorical variable with levels of HCL, Etomidate, Clove Oil, P/L, Vehicle, and MS222). Second, the number of entries in the lighted side was log-transformed to achieve a normal distribution of model residuals and homoscedasticity and regressed as a function of the fixed-effect treatment group (six-level categorical variable with levels of HCL, Etomidate, Clove Oil, P/L, Vehicle, and MS222) using the GLM procedure. To allow for the log-transformation, a constant 0.1 value was added to all data points to eliminate zero values. The number of days taken for the fish to learn to enter the lighted side was tested as a covariate in the model but removed due to a lack of significance.

For the cortisol analysis, the GLM procedure was used to test the cortisol level (response variable) as a function of the fixed-effect treatment group (six-level categorical variable as previously mentioned), fixed effect of sex (two-level categorical variable: female and male), fixed effect of time of sample collection after equilibrium loss (two-level categorical variable: 5 or 15 min), and the interaction between treatment group and sex.

## 3. Results

### 3.1. Animals Removed from the Analysis

Two animals from the P/L group and one animal from the etomidate group were not considered for the conditioned place avoidance analysis because they did not enter the lighted side during the exposure phase and thus did not have any contact with the conditioning agent. In addition, a total of nine animals were excluded from the cortisol analysis (two animals exposed for 5 min and four animals exposed for 15 min to etomidate; and one animal exposed for 5 min and two animals exposed for 15 min to P/L) because the cortisol levels in the biological samples from these animals were below the detection limit of the method used.

### 3.2. Conditioned Place Avoidance Test

#### 3.2.1. All Animals Took Longer to Go to the Lighted Side after Conditioning, Especially HCl Animals

Figure 3 illustrates the latency to enter the lighted side. This variable was significantly (*p* ≤ 0.0002) affected by the treatment group, phase, and the interaction between these two variables. Latency to enter the lighted side during the training baseline was considered a control; no differences were observed between groups (*p* > 0.05) for this phase. Regarding the exposure phase, MS222 animals were faster to enter the lighted side when compared with the latency of the HCL and the Etomidate groups (*p* ≤ 0.0194). Regarding the post-exposure phase, animals from the MS222, Clove Oil, P/L, and Vehicle groups had a lower latency to enter the lighted side than the HCL group (*p* ≤ 0.0060). Moreover, animals treated with etomidate took longer to enter the lighted side than the MS222 group (*p* = 0.0499). All groups took less time to enter the lighted side during the training baseline when compared with the exposure (*p* < 0.0001) or post-exposure (*p* < 0.0001) phases. Animals in the MS222, Vehicle, and HCl groups also showed an increased latency to enter in the lighted side in the post-exposure phase compared with the exposure phase (*p* = 0.0090, *p* = 0.0400, and *p* < 0.0001, respectively).

#### 3.2.2. Positive Control Group Animals Spent Less Time on the Lighted Side and Did Not Differ from the Etomidate Group

Figure 4 illustrates the time spent on the lighted side. This variable was significantly affected by the treatment group (*p* = 0.0154) and the interaction between the treatment group and phase (*p* = 0.0003). Regarding the pre-exposure phase, no differences between groups were observed for time spent in the lighted side. In the post-exposure phase, fish from the MS222, Clove Oil, P/L, and Vehicle groups spent more time on the lighted side than the HCL group (*p* ≤ 0.0022). In addition, fish from the HCl group spent less time on the lighted side during the post-exposure phase compared with the pre-exposure phase (*p* = 0.0164). The other groups spent similar times in the lighted side before and after conditioning.

#### 3.2.3. Positive Control Group Animals Entered the Lighted Side Less Often

Regarding the post-exposure phase, the number of entries into the lighted side was significantly affected by the treatment group (*p* < 0.0001). Here, fish from the MS222, Clove Oil, P/L, and Vehicle groups entered more often into the lighted side than the HCL group (*p* ≤ 0.0012) (Appendix A). The probability of an animal to enter the lighted side at least once was not affected by the treatment group. Nevertheless, all animals from the MS222, Vehicle, and P/L groups entered the lighted side after conditioning, while one animal from the Clove Oil group, three animals from the Etomidate group, and more than half the animals from the HCl group did not enter the lighted side (7/11).

### 3.3. Etomidate Animals Showed Lower Levels of Cortisol

For cortisol samples, sex was not found to have a significant effect. Time of collection after equilibrium loss (*p* = 0.0001) and interaction between this variable and treatment group (*p* = 0.0182) were significant, but no differences were found between treatment groups in samples collected 5 min after equilibrium loss (Figure 5). Regarding the samples collected 15 min after equilibrium loss, animals exposed to etomidate had significantly lower cortisol levels compared with the other groups (*p* ≤ 0.0001, *p* = 0.0018, and *p* = 0.0025 for the MS222, Clove Oil, and P/L groups, respectively) (Figure 5). The MS222 group was the only one with a significantly higher level of cortisol at 15 min when compared with the cortisol levels at 5 min (*p* = 0.0034) (Figure 5).

## 4. Discussion

Anaesthetic efficacy and potential side effects should be considered when selecting an anaesthetic protocol to ensure both animal welfare compliance and provision of scientifically sound data. One important aspect to consider is whether an animal perceives a drug as aversive, since this would alter the animal’s behaviour and physiology, potentially inducing distress and reducing welfare. Unfortunately, the most used anaesthetic, MS222, has been described as aversive to adult zebrafish [2,3], which motivated our search for better alternatives. The anaesthetic protocols tested in this study (MS222, propofol/lidocaine, clove oil, and etomidate) were not innocuous, as the animals subjected to these behaved differently before and after anaesthesia exposure. However, the MS222, P/L, and Clove Oil groups had a different behavioural profile compared with the positive control for aversion (HCl), contrary to etomidate. Regarding cortisol, all anaesthetic treatments except etomidate presented similar levels between them. The pH of the solution could contribute to the aversion levels, as seen in the HCl group (pH 3) where aversion was clearly present. Thus, we tested the remaining solutions and the pH influence was ruled out because they all had a pH around 7 (Table 1).

Contrary to what was found by Readman et al. [3], in the present study, animals exposed to etomidate showed some signs of aversion, whereas there was no clear aversion profile for MS222. These differences can probably be explained by the use of a different paradigm; in the study reported by Readman et al. [3], the animals moved freely between two sides, one containing only water and the other containing an anaesthetic solution of 50% of the effective anaesthetic concentration recommended [6]. In addition, the period of habituation to the apparatus was only 150 s, while in our experiment it was 48 h. Thus, our paradigm involved conditioned learning, while Readman et al. used a preference test, and the concentration we used was higher to mimic an experimental situation in which the animals had to be anaesthetised and may have felt aversion.

Our previous work [7] attempted to replicate the Wong et al. [2] experiment, but contrary to their results, animals treated with MS222 did not present a clear aversion. Animals were tested in pairs in the Wong et al. study, so the behaviour of one of the fish may have influenced the other, possibly altering the test results due to shoaling or territorial behaviours. Thus, in the present study, the animals were tested individually. During this period of short social isolation (13 days), the environment was enriched with a plastic plant [28]. In the present study, the results from clove oil-treated animals presented were similar to those reported by Wong et al., although they used a higher concentration. However, the MS222-treated animals in Wong et al. showed aversion by spending more time on the darkened side and avoiding the lighted side after conditioning, whereas our results showed no difference regarding this measure. In our study, the only group that significantly avoided the lighted side was the HCl group, which worked as the positive control for aversion. This control group showed that other dependent variables were relevant to measure aversion: the number of entries into the lighted side and the latency to enter into this side after conditioning; i.e., animals spent less time, exhibited fewer entries, and took more time to enter into the lighted side after being exposed to a pH of 3. Thus, these were the behavioural benchmarks presented by the animals that showed aversion to the anaesthetic/treatment.

Our previous studies [7] showed that animals subjected to propofol/lidocaine took more time to enter the compartment where the anaesthetics were presented than in the training baseline. We suspect that this may have been due to the white colour of propofol. In the present study, etomidate also had a white colour. To rule out the influence of the colour as a visual cue, a group of animals was exposed to Intralipid, the white vehicle of propofol and etomidate (Appendix A). When analysing the results, fish from the Vehicle group took longer to enter the lighted side after being exposed compared with the exposure phase. The P/L and Etomidate groups showed no differences in latency when comparing the exposure phase with the post-exposure phase. All these groups had Intralipid in the water during the post-exposure phase. These results pointed to the colour not being an aversive cue (lack of differences in latency between the exposure and post-exposure phases in P/L and Etomidate), but the re-exposure to the Intralipid alone may have been an aversive factor, which only occurred in the Vehicle group. This finding ruled out the possibility of visual cues skewing the results in the anaesthetic groups during the post-exposure phase. Nevertheless, in the exposure phase, all animals took more time to pass into the lighted side containing the treatment than in the training baseline, probably because they sensed the chemicals in the tube and/or, in the case of the P/L, Vehicle, and Etomidate groups, they observed a visual white cue. The strong odour from clove oil and the acidic properties and odour or taste of HCl [29] may also have been detected in the tube, thus increasing the latency to pass into the treated side during exposure. While the HCl animals took much more time to pass to the lighted side during the post-exposure phase than during the exposure phase, indicating a strong aversion, the same significant difference in the MS222 group may have been caused by the particularly low latencies presented during exposure.

Nevertheless, no anaesthetic protocol seemed to be completely innocuous, as all groups took more time to pass to the lighted side after being exposed. However, in the post-exposure phase, except for the Etomidate group, all the other groups exhibited significantly lower latencies to enter the lighted side compared with the ones from the positive control group. Thus, after a brief initial hesitation, the animals exposed to the anaesthetics (except to etomidate) behaved as if they found the conditioned side safe during the post-exposure phase. The HCl and Etomidate groups were found to spend less time and entered less often into the lighted side after conditioning, indicating an aversive profile for etomidate because it showed the same results as the positive control group.

It has been described that MS222 induced a long-term but not a short-term memory impairment in Medaka fish (*Oryzias latipes*) and rats [30,31], and it also induced working memory and cognitive flexibility impairment in adult zebrafish [32]; whereas etomidate induced amnesia [33], although no studies using zebrafish are known. Both lidocaine and propofol induced post-anaesthetic amnesia in other animal models, especially by impairing the consolidation of information retrieved immediately before or after the anaesthetic episode [34,35,36,37]. Contrary, clove oil has not been described to impair memory, and there are some studies related to memory improvement when animals were subjected to an amnesic [38]. Except for clove oil, the anaesthetics tested have been described to produce amnesia at some instance in different animal models. Thus, the lack of differences between the pre- and post-exposure phases regarding time spent and number of visits to the conditioning side could have been because the animals forgot the anaesthetic exposure. However, if the animals suffered from amnesia, they would have had similar latencies to enter into the lighted side before and after the conditioning, which did not occur. Nevertheless, in the case of the P/L group, the increase in latency can be explained by the presence of a visual cue on the lighted side during the post-exposure phase and not by this combination causing aversion. Therefore, the presence of an amnesiac effect could not be completely excluded. Although a visual cue was also present in the post-exposure phase for the Etomidate group, there were other dependent variables that showed an aversion degree similar to that of HCl.

Regarding cortisol, levels were not different between groups except for the Etomidate group, which presented lower cortisol levels at 15 min. This was expected because etomidate inhibits the pathway that leads to cortisol synthesis [22,39]; this can be problematic in several areas of research, so the anaesthetic selection must be carefully considered. Only the MS222-treated animals had higher cortisol levels at 15 min than at 5 min. Indeed, cortisol has been described to peak between 15 to 30 min after a stressful event in fish [37], and it is possible that the MS222-treated animals had a quicker cortisol release than the other groups. The values presented here at 15 min (except for the Etomidate group, as explained) were higher than what we found as baseline cortisol levels in animals euthanised with rapid cooling (mean ± SD of 11.52 ± 5.49 ng/g) [40]. Thus, anaesthesia induction may be a stressful event, as also shown by the behavioural data in which the anaesthetic exposure was not innocuous. We do not know if the stress was related to the chemical compounds per se or if it was behaviourally mediated. The latter explanation seems more likely because the chemical compositions of these anaesthetics are very different and the animals presented high levels of cortisol in all the groups (except the Etomidate group) when compared with the baseline reference.

## 5. Conclusions

In sum, the stress of anaesthesia induction as measured according to cortisol levels was similar between groups and higher than the baseline reference except for the Etomidate group. This group showed a behavioural profile similar to the positive control group (HCl) in the CPA test, thus presenting some degree of aversion; this did not contradict the low level of cortisol detected, as this result may have been due to the pharmacological effect of etomidate inhibiting cortisol synthesis. Nevertheless, it cannot be assured that any of these anaesthetics are innocuous and do not cause aversion in adult zebrafish, as the latencies to enter the lighted side were altered after exposure to these agents. It is possible that the loss of control of the body during anaesthesia induction could be enough to elicit aversion and a stress response. Except for Etomidate group, the other variables measured to detect aversion did not change after conditioning. Therefore, future studies using different behavioural paradigms that assess brain activity will be important to clarify anaesthesia aversion. In addition, it will also be important to clarify the post-anaesthetic amnesia using these anaesthetic protocols.

In conclusion, according to our results, we cannot recommend the use of the etomidate protocol to avoid aversive responses, but all the other anaesthetic protocols (propofol with lidocaine, clove oil, and MS222) seemed to be equally valid regarding the avoidance of aversion and stress.

## Figures and Tables

**Figure 1 biology-11-01433-f001:**
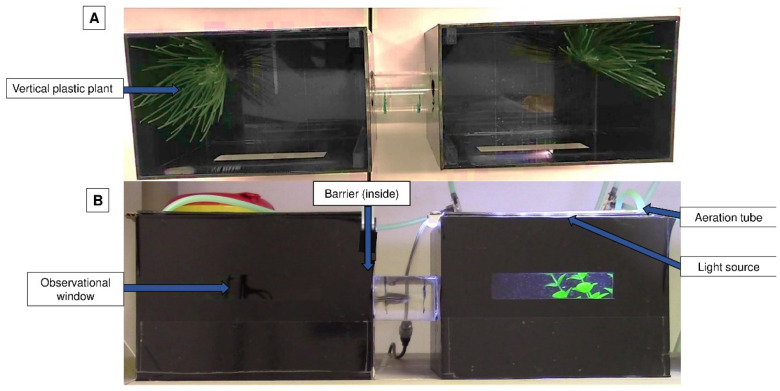
Conditioned place aversion apparatus view from above (**A**) and from the side (**B**). Tanks were individually lighted and only one tank had the lights on at a certain time. A transparent tube linked the tanks for the animals to pass; a plant and aeration were provided on each side to increase animal welfare.

**Figure 2 biology-11-01433-f002:**
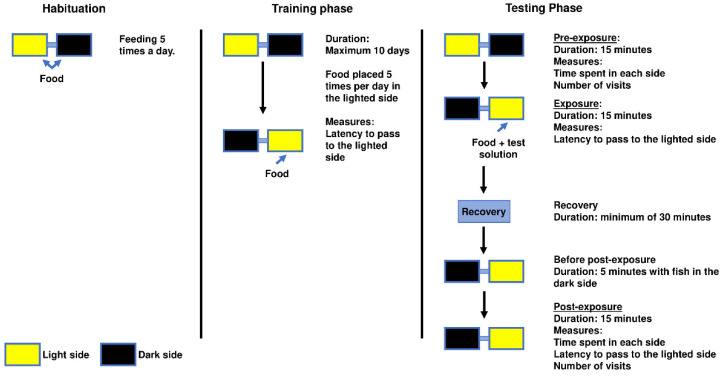
Conditioned place aversion behaviour test scheme with measurements taken in each phase. The habituation phase lasted for 48 h followed by a training phase in which the training baseline was established; training was completed when the criteria for fully trained animals was reached (see Section 2). Afterwards, the testing phase was divided into pre-exposure and post-exposure performed on the same day. Recovery represents the recovery of the anaesthesia or other testing solution when the animal was placed in clean water. Which tank was lighted at each given time and where the food and testing solutions were introduced are also depicted.

**Figure 3 biology-11-01433-f003:**
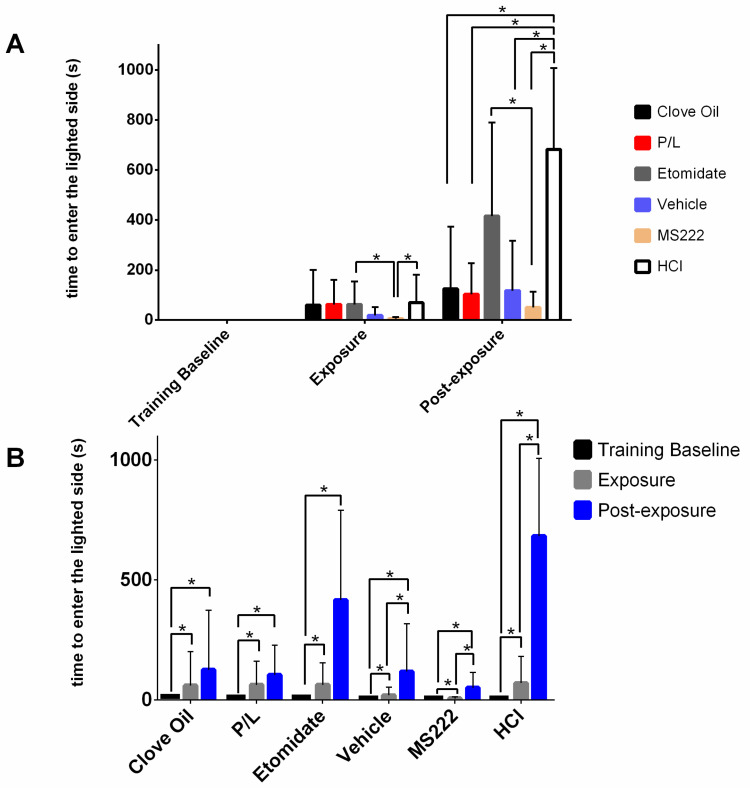
Observed median latency to enter the lighted side of the apparatus and their respective interquartile ranges on the last day of training (Training Baseline) in the exposure phase and in the post-exposure phase for each treatment group. Clove Oil group: animals subjected to 45 mg/L of clove oil (n = 12); P/L group: animals subjected to 5 mg/L of propofol combined with 150 mg/L of lidocaine (n = 10); Etomidate group: animals subjected to 2 mg/L of etomidate (n = 9); Vehicle group: animals subjected to 5 mg/L of intralipid (n = 12); MS222 group: animals subjected to 150 mg/L of MS222 (n = 12); HCl group: animals subjected to pH 3 water with hydrochloride acid (n = 11). Animals that did not enter into the lighted side had an attributed latency of 900 s. (**A**) Illustration of the comparisons between groups within each testing phase; * *p* < 0.019. (**B**) Illustration of the differences between each phase for each experimental group; * *p* < 0.040. The analysis was performed using the GLIMMIX procedure with a log-normal response distribution; asterisks indicate significant differences between least-square means. The training baseline latencies were so low that they are not visible in panel A; the corresponding bars were made visible in panel B to facilitate the figure legibility (Clove Oil: 7.5 (7.3) s, P/L: 6.0 (9.8) s, Etomidate: 9.0 (2.3) s, Vehicle: 8.5 (10.8) s, MS222: 9.0 (7.3) s, HCl: 6.0 (7.0) s).

**Figure 4 biology-11-01433-f004:**
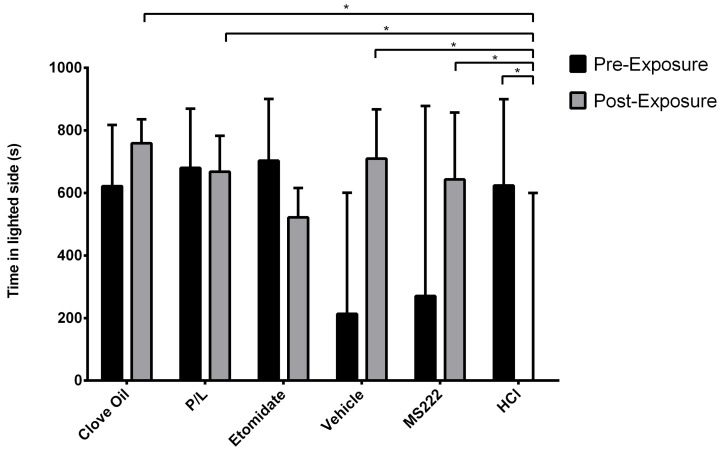
Observed median time spent on the lighted side of the apparatus and their respective interquartile ranges in each treatment group during the pre-exposure and post-exposure phases. Clove Oil group: animals subjected to 45 mg/L of clove oil (n = 12); P/L group: animals subjected to 5 mg/L of propofol combined with 150 mg/L of lidocaine (n = 10); Etomidate group: animals subjected to 2 mg/L of etomidate (n = 9); Vehicle group: animals subjected to 5 mg/L of intralipid (n = 12); MS222 group: animals subjected to 150 mg/L of MS222 (n = 12); HCl group: animals subjected to pH 3 water with hydrochloride acid (n = 11). * *p* ≤ 0.016 using the MIXED procedure for testing the square-root-transformed lighted-side index (time spent on the lighted side/total time spent in both sides); asterisks indicate significant differences between least-square means of groups and phases.

**Figure 5 biology-11-01433-f005:**
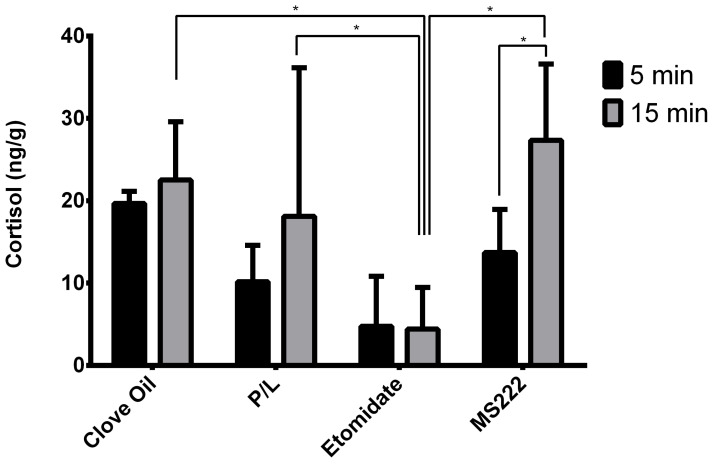
Observed medians of trunk cortisol levels 5 min or 15 min after zebrafish had lost their equilibrium and their interquartile ranges. Clove Oil group: animals subjected to 45 mg/L of clove oil (n = 12); P/L group: animals subjected to 5 mg/L of propofol combined with 150 mg/L of lidocaine (n = 11); Etomidate group: animals subjected to 2 mg/L of etomidate (n = 10); MS222 group: animals subjected to 150 mg/L of MS222 (n = 12). * *p* ≤ 0.003 significant differences between least-square means of groups and phases using the GLM procedure.

**Table 1 biology-11-01433-t001:** The pH of the solutions and the system water used in the study as measured by a pH probe (Inolab, level 1, Xylem Analytics Germany Sales GmbH & Co. KG, Oberbayern, Germany). The solution temperature was 26 ± 1 °C.

	Concentration	pH
MS222	150 mg/L	7.05
Etomidate	2 mg/L	6.97
Vehicle for etomidate	60 mg/L Intralipid	6.86
Clove oil	45 mg/L	7.15
Propofol/lidocaine	5 mg/L propofol + 150 mg/L lidocaine	7.05
Vehicle for propofol/lidocaine	5 mg/L Intralipid	6.93
Vehicle	5 mg/L Intralipid	6.93
HCl	4.67 mg/L	3.00
System water		6.94

## Data Availability

All relevant data are within the manuscript and the Appendix A.

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
