# Peer review of "Behavioural Aversion and Cortisol Level Assessment When Adult Zebrafish Are Exposed to Different Anaesthetics"

_biology, 2022, doi:10.3390/biology11101433_

Round 1

Reviewer 1 Report

In the manuscript, Biology-188355 by Ferreira et al, authors examined different anesthetics in Zebrafish for their effect on behavioral aversion and cortisol levels. A combination of propofol with lidocaine, clove oil, and etomidate were used in this study. Upon assessment of the aversion, the authors found that except for etomidate, other anesthetics were valid to be used in laboratory settings. The use of appropriate anesthetics is important for the in vivo studies in Zebrafish; hence assessment of suitable chemicals is warranted. Taking this into consideration, the study is interesting. In general, the manuscript is well-written. The conclusions made here are in line with the data presented. However, there are some suggestions/comments to improve the manuscript.

1.    Abstract. The very sentence can be revised. "is a popular animal model" can be replaced with a meaningful adjective related to it types of studies it is used for.

2.    Table 1. what was the basis for choosing these particular concentrations?

3.    Legends of Figures 1 and 2 can be a little more descriptive.

4.    Result section. I would recommend revising the result section. In the current form, the result headings are not informative. It appears that the authors have used the test/subject as a heading and then described the results in text content. It would be more easy following if the authors use the conclusion sentences as the heading in the results.

5.    What is the conclusion of the result heading 3.1? In the present form, the “Animals” is not informative.

6.    Heading 3.2 says “Conditioned place Avoidance test” and one bold line. For the readers not well versed in the system, it would not be easy to follow up.

7.    Similarly heading 3.3., “Cortisol analysis”. This heading sounds like a test. Please use the concluding sentence or message of the data presented.

8.    In figures 3, 4, and 5, Please define the a, b, AB, and other letters in their legends.

Author Response

The authors would like to thank you very much for all the input and comments.

Please see the attachment for the answer addressing all of them.

Reviewer 2 Report

In the present study, authors have studied the effect of different anesthetics on the aversion-associated behaviors of the adult zebrafish by using several anesthetics. The experiments were well designed and performed with the proper controls. Overall the study is good but I have some serious concerns over the representation of the data and statistics.1- In figure 3- since the error bars in each condition ( Exposure and Postexposure) are showing huge variability, then how come there is a significant difference in the group? it should be clarified what groups are having significant differences with which group in the text correlating with the letters.  2- Figure:4- The number of entries in the lighted side should also be plotted between Pre and Post-conditioning, as it can be a good measure of the anxiety. 

Author Response

(The authors gave the same response as above.)

Reviewer 3 Report

The study performed by Ferreira et al. raises important issues because use of proper anesthesia in zebrafish research is essential to ensure fish welfare and data reliability.

The introduction and discussion section are well presented.

The issues that have to be corrected/explained:

1.    The authors created vehicle group to rule out the influence of white color of propofol and etomidate. What about another groups? What is the color of the solutions? If not white, proper control should be also added.

2.    The presentation of graphs and statistic has to be improved. Firstly, it is not clear if the authors compared each group with each other or with some reference group? Next, Figure 3 – if training baseline is 0 for each exposure group it should be presented as line not empty space without any bars. The letter markings are very confusing. For example: the difference between vehicle and ms222 in exposure phase is very clear and the use of A and B is relevant. However the use of B in clove oil and P/L is confusing. Are the values really the same as in ms222 group? The lower-case letters are also illogically used. Figure 4 – to what value the AB letter refers to? Post-exposure etomidate is significantly different in comparison to which group? And again the lower-case letters are illogically used. In both, Fig 3 and 4 the errors bars are big, sometimes higher than the bar. Please reconsider the basic statistic and the gaussian distribution. In Figure 5 there is an evident lack of control group.

Author Response

(The authors gave the same response as above.)

Round 2

Reviewer 2 Report

The manuscript is much improved now. The manuscript should be accepted in its current form.